# Neuromodulation Therapy for Chemotherapy-Induced Peripheral Neuropathy: A Systematic Review

**DOI:** 10.3390/biomedicines10081909

**Published:** 2022-08-07

**Authors:** Ryan S. D’Souza, Yeng F. Her, Max Y. Jin, Mahmoud Morsi, Alaa Abd-Elsayed

**Affiliations:** 1Department of Anesthesiology and Perioperative Medicine, Division of Pain Medicine, Mayo Clinic, Rochester, MN 55905, USA; 2Department of Anesthesiology, University of Wisconsin, Madison, WI 53706, USA; 3Department of Anesthesiology, John H. Stroger, Jr. Hospital of Cook County, Chicago, IL 60621, USA

**Keywords:** spinal cord stimulation, dorsal root ganglion stimulation, peripheral nerve stimulation, chemotherapy, cancer pain, peripheral neuropathy, clinical outcomes, chronic pain

## Abstract

Chemotherapy-induced peripheral neuropathy (CIPN) is a debilitating and painful condition in patients who have received chemotherapy. The role of neuromodulation therapy in treating pain and improving neurological function in CIPN remains unclear and warrants evidence appraisal. In compliance with the Preferred Reporting Items for Systematic Reviews and Meta-Analyses (PRISMA) guidelines, we performed a systematic review to assess change in pain intensity and neurological function after implementation of any neuromodulation intervention for CIPN. Neuromodulation interventions consisted of dorsal column spinal cord stimulation (SCS), dorsal root ganglion stimulation (DRG-S), or peripheral nerve stimulation (PNS). In total, 15 studies utilized SCS (16 participants), 7 studies utilized DRG-S (7 participants), and 1 study utilized PNS (50 participants). Per the Grading of Recommendations, Assessment, Development, and Evaluations (GRADE) criteria, there was very low-quality GRADE evidence supporting that dorsal column SCS, DRG-S, and PNS are associated with a reduction in pain severity from CIPN. Results on changes in neurological function remained equivocal due to mixed study findings on thermal sensory thresholds and touch sensation or discrimination. Future prospective, well-powered, and comparative studies assessing neuromodulation for CIPN are warranted.

## 1. Introduction

Chemotherapy-induced peripheral neuropathy (CIPN) is a common and debilitating condition in cancer patients who have received chemotherapy. CIPN is predominantly a sensory neuropathy localized in a stocking-glove distribution. Sensory symptoms may range from allodynia to hypoesthesia and may be accompanied by motor and autonomic deficits [1], resulting in increased fall risk and decreased quality of life [2,3]. The severity of CIPN is dependent on the type of chemotherapy agent and its associated neurotoxicity profile, cumulative dose, and duration of exposure [4,5,6]. Symptoms generally improve with discontinuation of the chemotherapy agent over time, although complete recovery to baseline is not always attained. It is estimated that 50–90% of patients who receive chemotherapy go on to develop CIPN in the acute setting, and 30–40% of patients continue to suffer from CIPN chronically [1,4,6].

Current pharmacologic therapy for CIPN includes a multimodal strategy consisting of the following regimen: acetaminophen, non-steroidal anti-inflammatory drugs (NSAIDs), topical agents (e.g., lidocaine and capsaicin), anti-convulsants (e.g., gabapentinoid and carbamazepine), anti-depressants (e.g., selective norepinephrine reuptake inhibitors and tricyclic anti-depressants), opioids, and cannabinoids [1,7]. However, their effectiveness may be limited. Low therapeutic index and adverse effect profiles from some medications may present additional challenges. Neuromodulation therapy has emerged as a viable treatment option for many refractory cases of chronic pain, including CIPN [8].

This review focuses on dorsal column spinal cord stimulation (SCS), dorsal root ganglion stimulation (DRG-S), and peripheral nerve stimulation (PNS). Dorsal column SCS involves the placement of leads in the epidural space overlying the dorsal column of the spinal cord [8]. Various stimulation waveforms, including tonic stimulation, high-frequency (10-kHz) stimulation, burst stimulation, differential target multiplexed stimulation, and closed-loop stimulation, can be delivered via dorsal column SCS [9,10,11]. DRG-S involves lead placement in the epidural space overlying the dorsal root ganglion [12,13]. PNS involves lead placement in close proximity to a target nerve that innervates the location of the painful area [14,15].

Evidence from landmark trials has highlighted the efficacy of SCS, DRG-S, and PNS in the treatment of neuropathic pain conditions, revealing superior pain relief, physical functionality, mental health, and patient satisfaction compared to conventional medical management alone [10,16,17,18,19]. Studies have also highlighted that animal models with CIPN may benefit from stimulation. For instance, a study on rats with paclitaxel-induced mechanical and cold hypersensitivity revealed that SCS therapy significantly inhibited the development of mechanical and cold hypersensitivity compared to rats receiving paclitaxel alone or sham SCS [20]. To date, there is Food and Drug Administration (FDA) approval supporting SCS therapy for painful diabetic neuropathy (PDN) [21,22], failed back surgery syndrome [23], non-surgical refractory back pain [24], and complex regional pain syndrome (CRPS) [25]. DRG-S is approved for use in CRPS, while PNS is approved for the treatment of chronic pain, post-surgical pain, or post-traumatic pain located in the back, extremities, or head/neck.

In this systematic review, we appraise the current literature on the use of SCS, DRG-S, and PNS for the treatment of pain related to CIPN. The primary objective is to assess changes in pain intensity. The secondary objective is to assess changes in neurological function.

## 2. Methods

### 2.1. Compliance with Ethics Guidelines

This article is based on previously conducted studies and does not contain any new studies with human participants or animals performed by any of the authors. The protocol was registered in the International Prospective Register of Systematic Reviews (PROSPERO CRD42022338500).

### 2.2. Search Strategy

This study followed the Preferred Reporting Items for Systematic Reviews and Meta-Analysis (PRISMA) guidelines [26]. As such, a comprehensive search of several databases from each database’s inception to 16 May 2022 in the English language was conducted, along with the creation of a hand-searched reference list of identified publications. The databases included Ovid MEDLINE(R), Ovid EMBASE, Ovid Cochrane Central Register of Controlled Trials, Ovid Cochrane Database of Systematic Reviews, and Scopus. The search strategy was designed and conducted by an experienced librarian (L.J.P.) with input from the study’s principal investigator (R.S.D.). Controlled vocabulary supplemented with keywords was used to search spinal cord stimulation and peripheral nerve stimulation for CIPN in humans. The actual strategy listing all search terms used and how they are combined is available in the Appendix A.

### 2.3. Study Selection

Included studies abided by the following criteria: any study design that involved humans undergoing neuromodulation (SCS, DRG-S, PNS) for the treatment of CIPN. Exclusion criteria consisted of the following: review or meta-analysis articles. Of note, we included abstracts in this review. Two authors (M.Y.J. and M.M.) independently selected abstracts along with full-text articles from the above-listed databases, while a third author (R.S.D.) resolved any discrepancies.

### 2.4. Data Extraction

The following data on the characteristics of each study were extracted: study design, funding source, mean age and number of participants, type of intervention and lead placement location, and stimulation settings. Relevant data on the primary and secondary outcomes of interest were also extracted. The primary outcome of interest was participant-reported changes in pain intensity related to their CIPN after neuromodulation therapy. Secondary outcomes of interest included participant-reported changes in neurological function (e.g., change in motor or sensory deficit). For each included study, two authors (M.Y.J. and M.M.) independently extracted all relevant data, with a third author (R.S.D.) arbitrating any disputes.

### 2.5. Assessment of Risk of Bias

Two independent reviewers (Y.F.H. and R.S.D.) assessed the risk of bias for the included studies using the Newcastle–Ottawa Quality Assessment Scale for case-control and cohort studies. There was no randomized controlled trial captured in this review. For observational studies, bias was assessed based on the following domains: Selection (representativeness of the exposed cohort, selection of the non-exposed cohort, ascertainment of exposure, demonstration that outcome of interest was not present at the start); Comparability (comparability of cohorts on the basis of the design or analysis); and Exposure/Outcome (assessment of outcome, follow-up long enough for outcomes to occur, adequacy of follow-up of cohorts). A maximum of one star for each numbered item within the Selection domain and Exposure/Outcome domain can be awarded per study. A maximum of two stars can be awarded for the Comparability domain. A higher number of stars indicates a lower risk of bias for each respective domain.

### 2.6. Quality of Evidence

GRADEpro software (McMaster University and Evidence Prime, Inc., Kraków, Poland, 2022; http://gradepro.org, accessed on 21 June 2022) was used to assess the quality of evidence for each outcome per GRADE (Grading of Recommendations, Assessment, Development, and Evaluations) criteria. Randomized controlled trials are categorized as high-level evidence. This can be downgraded based on the risk of bias, inconsistency, indirectness, imprecision, and publication bias.

## 3. Results

### 3.1. Search Strategy and Study Selection

The search strategy identified 847 citations. After independent and duplicate screening, 40 full-text articles were assessed, and 23 articles were selected that met the full inclusion criteria. The full study selection process is presented in Figure 1. All included studies were case reports/series except for one retrospective observational study [27]. In total, the studies comprised 73 total participants. Of these, 15 studies [28,29,30,31,32,33,34,35,36,37,38,39,40,41,42] utilized SCS (*n* = 16 participants), 7 studies [43,44,45,46,47,48,49] utilized DRG-S (*n* = 7 participants), and 1 study [27] utilized PNS (*n* = 50 participants).

### 3.2. Evidence Supporting Neuromodulation Therapy for Pain Relief

SCS therapy for CIPN-related pain was evaluated across 15 case studies [28,29,30,31,32,33,34,35,36,37,38,39,40,41,42] and reported in Table 1. A total of 3 recent case reports reported improvement in pain intensity by 80% during the trial phase, which was maintained after permanent SCS implantation, with visual analog scale (VAS) scores ranging between 0/10 and 2/10 at 2 years follow-up [28,35,36]. Cata et al. [41] reported SCS implantation in two patients with CIPN. Of these, 1 patient reported a VAS score of 4.5/10 at baseline, which improved to 0/10 after SCS therapy, and substantial pain relief was maintained at 4 months follow-up. The second patient had initial improvement of pain from 4.6/10 at baseline to 0/10 during the trial period and after implantation but reported a slight increase in VAS score to 3.6/10 at the 2-week follow-up. Regardless, the patient reports satisfaction with SCS therapy and improvement in gait and leg flexibility [41]. Panchal et al. reported the use of a wireless SCS device that provided more than 90% improvement in pain intensity from CIPN [33].

DRG-S for CIPN-related pain was evaluated in seven case reports [43,44,45,46,47,48,49]. Of these, 3 recent case studies reported 100% improvement in pain with VAS scores ranging between 0/10 to 1/10, and this substantial relief was maintained between 5 months to 3 years post-implant [44,48,49]. Finney et al. and Yelle et al. reported 50% and 60% improvement in pain intensity, respectively, after DRG-S [46,47]. However, notable complications have been reported with DRG-S use. For example, Sindhi et al. [43] described a male patient who suffered sharp stabbing 10/10 pain intensity at the glans of the penis after penile mass excision and chemoradiotherapy treatment. The patient reported >60% improvement in pain after DRG-S. However, after 5 months, there was a severe increase in pain which was not responding to reprogramming, which was attributed to slight dorsal migration of the DRG leads. Similarly, Rao et al. [45] reported >75% improvement in pain with the DRG-S trial, although immediately after permanent implantation, the patient suffered sudden lower extremity numbness. This was attributed to the right DRG lead traversing the dura anterior to the spinal cord and compressing the right S1 nerve root before exiting the thecal sac. This lead was removed immediately, and the patient underwent re-implantation one month later with adequate coverage of the right posterior calf, although with persistent right foot pain.

The impact of PNS in alleviating neuropathic pain was evaluated in 1 retrospective study of 50 participants. Sacco et al. [27] performed percutaneous auricular neurostimulation (PANS) therapy which stimulated peripheral nerves in the ear. All 18 participants who were available for quantitative assessment of pain intensity reported improvement in pain VAS scores from a mean of 8.11 to 3.77. Of the remaining 32 participants available for qualitative pain assessment, 59% reported marked improvement, 12.5% had minimal improvement, and 29% reported no improvement.

### 3.3. Evidence Supporting Neuromodulation Therapy for Neuropathy and Neurological Deficits

The role of SCS in improving neuropathy and neurological deficits from CIPN was reported in eight case reports [28,33,35,36,39,40,41,42] consisting of nine patients (Table 1). Mixed results were reported regarding temperature sensation. Kamdar et al. [35] reported a reduction in cold hypersensitivity after SCS therapy. However, Cata et al. [41] reported no change in thermal thresholds after SCS therapy for one of their evaluated patients. Outcomes for touch sensation also revealed equivocal and mixed results. Kamdar et al. [35] reported an unchanged sensory exam post-implant, while Cata et al. [41] reported improved touch detection in both of their patients after SCS therapy. Furthermore, Cata et al. reported that sharpness detection was only improved in one of the two patients [41]. Six studies consisting of seven participants reported improved ability to participate in daily activities and more mobility with reasons attributed to improved gait [35,41], more stability [35], better flexibility of legs [41], ambulation with less pain [33], decreased dependence on assistance from others [39], and improvement of hand function [36,42].

Change in neurological function following DRG-S was evaluated in six studies [43,46,48] consisting of six patients. Two studies reported results on touch sensation, with one study [48] citing no change in sensation to light touch or pinprick sensation (*n* = 1), while another study [45] reporting worsened lower extremity numbness after DRG-S. Four studies consisting of four participants reported improved ability to participate in daily activities and more mobility, with reasons being attributed to improved ability to wear shoes and exercise [44,48] and reduced pain while working [43] or walking [43,46].

The impact of PNS on neurological deficits was evaluated in 1 retrospective study of 50 participants [27]. In this study utilizing PANS therapy, participants reported notable qualitative improvements in numbness, gait, and balance. One patient reported adverse outcomes due to the intolerance of intermittent pulsing.

### 3.4. Bias Assessment

The risk of bias was assessed for one observational study [27]. Since the remaining studies were case reports/series, they were considered high risk for bias and not amenable to appraisal with bias assessment tools. Using the Newcastle–Ottawa Quality Assessment Scale, the study by Sacco et al. [27] received two stars for the Selection domain (maximum possible four stars) and two stars for the Exposure/Outcome domain (maximum possible three stars). There were no stars assigned to the Comparability domain, indicating a high risk of bias for that respective domain.

### 3.5. Quality of Evidence

An evidence profile table and summary of findings table with quality of evidence for each outcome are summarized in Table 2 and Table 3. Overall, there was very low quality of evidence that neuromodulation interventions (SCS, DRG-S, PNS) were effective in reducing pain intensity from CIPN. Results were equivocal regarding the role of neuromodulation interventions in improving neurological function, with some reporting improvements, some reporting no change, and some reporting deteriorating neurological deficits. For both outcomes, there was a high risk of bias, inconsistency, indirectness, impression, and potential publication bias in the included studies. A total of 22 studies were case reports, and 12 studies were non-peer-reviewed abstracts presented at conferences.

## 4. Discussion

### 4.1. Summary of Evidence

This systematic review appraises the evidence of neuromodulation therapy for pain and neuropathy in CIPN. Overall, this review highlights that dorsal column SCS, DRG-S, and PNS are associated with a moderate to high reduction in pain severity from CIPN. Pain relief from dorsal column SCS ranged from 50–100% between 3 months and 2 years, whereas pain relief from DRG-S ranged from 50–100% between 1 month and 3 years. These findings are concordant with the literature highlighting the efficacy of neuromodulation therapy for a variety of neuropathic pain conditions, including failed back surgery syndrome [50] and PDN [51]. Furthermore, although the mechanism of nerve injury between CIPN and PDN differs, they both present similarly with distal axonopathy and symmetrical length-dependent sensory neuropathy in a stocking-glove distribution [52]. Thus, it would seem rational that the response to neuromodulation would be similar between CIPN and PDN.

Efficacy of PNS therapy was highlighted in 50 participants, with reports of >50% pain relief among those who reported quantitative pain assessments. However, in this study [27], it is unclear how auricular nerve stimulation may alleviate CIPN which typically manifests in the distal extremities in a stocking-glove distribution. PNS implantation at the site of CIPN (e.g., lower extremity peripheral nerves or upper extremity peripheral nerves) was not described. Due to the very low-quality GRADE evidence supporting decreased pain intensity from neuromodulation interventions, additional prospective and powered trials are warranted. There are ongoing prospective clinical trials assessing the efficacy of dorsal column SCS for CIPN (ClinicalTrials.gov Identifier: NCT05411523).

In terms of change in neurological function and neuropathy assessment metrics, results from studies remained equivocal. Although most studies reported improvement in numbness, gait, balance, and ambulation, results were mixed with regard to thermal sensory thresholds, touch sensation, and touch discrimination. Similar to the primary outcome of pain intensity, there was very low-quality GRADE evidence on the outcome of neurological function.

Given the stocking-glove distribution of CIPN which frequently involves multiple nerve distributions, the authors recommend offering dorsal column SCS as the neuromodulation intervention of choice due to its ability to target broader areas of pain. DRG-S also has the capability of providing alleviation across multiple nerve distributions, partly due to its mechanism of cross-talk between dorsal root ganglions at different levels [13]. PNS may be considered if the patient’s pain symptoms are primarily located in one to two nerve distributions. The decision to pursue a specific neuromodulation intervention should also weigh its respective adverse effect profile. Overall, studies have highlighted that SCS [53,54,55], DRG-S [56], and PNS [15] are safe interventions overall. Special considerations unique to patients with CIPN may include an immunocompromised status due to ongoing malignancy and/or chemotherapy, which is associated with an increased risk for infection. Coagulopathy due to the underlying malignancy or due to side effects from chemotherapy may increase the risk for hematoma, which is particularly worrisome in the setting of neuraxial hematoma from SCS or DRG-S placement.

Our review suggests that clinicians caring for patients with CIPN should consider neuromodulation options in their treatment algorithm, particularly due to the paucity of evidence supporting conservative measures and conventional pharmacotherapy. Conventional treatment options for CIPN are highlighted by the American Society of Clinical Oncology (ASCO) guidelines [57]. The ASCO guidelines confirmed that no pharmacotherapeutic agents are recommended for CIPN prevention. Certain strategies such as dose delaying, dose reduction, substitution, and stoppage of chemotherapy may be considered in patients who develop intolerable symptoms from CIPN. In terms of treatment after the occurrence of CIPN, duloxetine is the only current neuropathic agent with intermediate-level evidence to support use in CIPN with moderate-level efficacy. Importantly, the ASCO guidelines recommended against the use of other commonly used neuropathic agents such as amitriptyline, gabapentin, pregabalin, and venlafaxine due to no evidence of efficacy or low-level efficacy. Importantly, many neuropathic medications may have a high prevalence of side effects such as somnolence, dizziness, difficulty in concentration, and other drug-specific side effects [7]. Further evidence is warranted for other conservative treatment options such as scrambler therapy, acupuncture, and exercise [57].

In summary, prior to offering neuromodulation options for the treatment of pain in CIPN, the authors recommend offering first-line conventional therapy such as multimodal pharmacologic therapy and physical therapy. Although pharmacologic therapy may consist of non-steroidal anti-inflammatory drugs, gabapentinoids, and anti-depressants, the literature supports that only duloxetine should be offered for treatment due to intermediate-level evidence for moderate efficacy. Of note, CIPN may also improve over time after completion of chemotherapy. In the setting of conventional treatment failure, pain specialists may offer neuromodulation options, including SCS, DRG-S, and PNS.

### 4.2. Proposed Mechanism of Action

Several mechanisms of action have been proposed to explain the improvement of pain from neuromodulation. Although a commonly proposed mechanism is the gate control theory, new mechanisms have been proposed associated with unique waveform paradigms, including high-frequency, burst, and other waveforms [58]. In high-frequency SCS, the gate control mechanism may be activated without stimulating pathways for paresthesia [59]. In burst stimulation, both the lateral and medial aspects of the spinothalamic tract are activated [60]. Besides the gate control theory, analgesia from SCS may also be explained by its modulation of neurotransmitters (cholinergic, serotonergic, and opioidergic pathways), depression of wide dynamic range neurons, and activation of supraspinal levels from orthodromic dorsal column action potentials [61]. In PNS, modulation of the central nervous system involves the dorsal lateral prefrontal cortex, parahippocampal areas, and the anterior cingulate cortex [14,62].

The mechanisms explaining the improvement of neurological deficits from neuromodulation therapy remain unknown. A prior review by Khunda et al. proposed that the neurological improvement of neurogenic urinary and bowel disorders through neuromodulation may be due to the stimulation of afferent pathways from the genital area [63]. Recently, a study revealed that activity-dependent SCS could rapidly restore truncal and lower extremity motor function in patients with complete paralysis [64]. Future studies are warranted to elucidate the mechanisms of improvement in neurological function after neuromodulation therapy.

### 4.3. Limitations

Clinical heterogeneity was significant across studies that evaluated different neuromodulation interventions for CIPN. Even in studies that focused on one neuromodulation intervention (e.g., DRG-S), there was variable placement of lead location or variable waveform parameters. Furthermore, details regarding waveform settings were not provided in several studies. Due to this level of heterogeneity, outcome measurements were unable to be pooled. Only one observational study was captured, and the remaining evidence was limited to case series/reports. Although this review evaluated all neuromodulation interventions, the majority of studies focused on dorsal column SCS and DRG-S. Another reason for low-quality evidence was the absence of placebo-controlled randomized controlled trials. A prior review of placebo-controlled trials highlights the improvement in neuropathic pain syndromes from neuromodulation interventions [50]. However, challenges to performing placebo-controlled studies in neuromodulation are prominent and include difficulty in developing placebo or sham arms, issues with inadequate blinding, placebo and nocebo effects, and, most importantly, ethical concerns centered over the concept of equipoise [65]. Thus, with these significant limitations and very low-quality GRADE evidence, it is imperative that the findings from this systematic review be interpreted with caution. Clinical implications from this systematic review, whether in support of or against the efficacy of neuromodulation for improving pain and neurological function from CIPN, cannot be concretely concluded.

### 4.4. Future Directions

Future studies should conduct adequately powered and prospective studies assessing neuromodulation for CIPN. Although efficacy may be dependent on the type of waveform, as demonstrated in prior studies [66,67], included studies did not compare the efficacy between different waveforms for CIPN. Thus, comparative studies assessing different waveforms are warranted. Finally, primary care clinicians, oncologists, and hematologists rarely consider neuromodulation for the treatment of cancer-related pain. Therefore, dissemination of information and education for both physicians and patients via conference proceedings, social media coverage [68,69], and other avenues of information dissemination are important [70].

## 5. Conclusions

Our systematic review performed an evidence synthesis on neuromodulation interventions for the treatment of CIPN. The current evidence suggests that neuromodulation interventions, including SCS, DRG-S, and PNS, may lead to clinically meaningful pain relief in patients with CIPN. Improvement in gait, motor function, and sensory function was also highlighted in several studies. The GRADE certainty of evidence is limited due to a lack of prospective comparative studies, clinical and methodological heterogeneity, and low sample size.

## Figures and Tables

**Figure 1 biomedicines-10-01909-f001:**
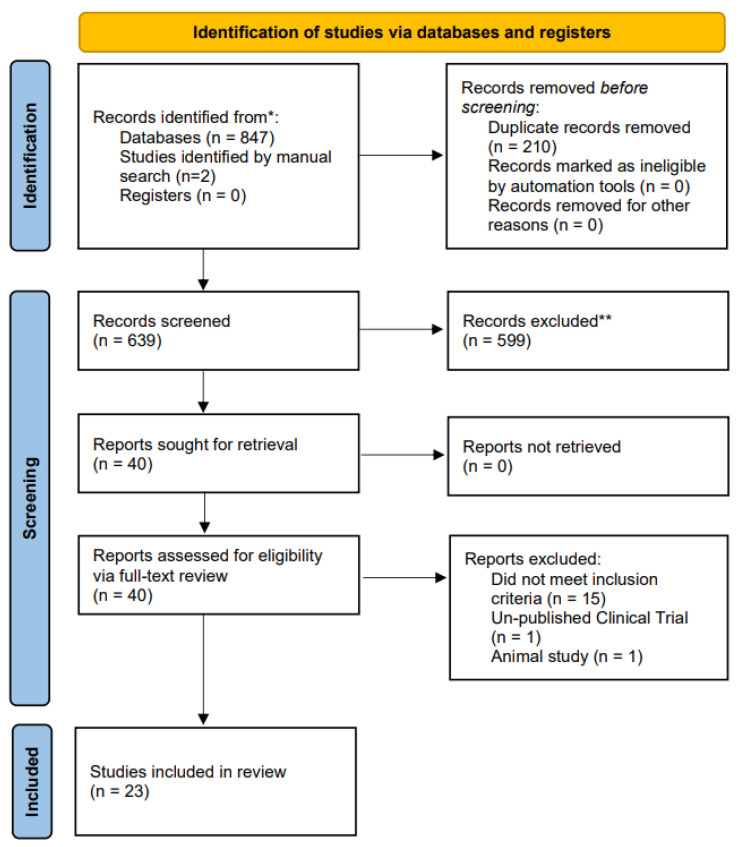
PRISMA flow diagram for systematic review. Flow chart of the study selection process, inclusion and exclusion of studies, and reasons for exclusion are displayed. * Databases included Ovid MEDLINE(R), Ovid EMBASE, Ovid Cochrane Central Register of Controlled Trials, Ovid Cochrane Database of Systematic Reviews, and Scopus. ** Records excluded from title and abstract screening only.

**Table 1 biomedicines-10-01909-t001:** Summary of Studies on Neuromodulation Interventions for Chemotherapy-Induced Peripheral Neuropathy.

Author/ Year	Study Design	Funding Source	Mean Age	Type of Intervention (Location)	Stimulation Settings	Pain Outcomes	Neurological Function Outcomes	Other Outcomes
**Dorsal Column Spinal Cord Stimulation**								
Abd-Elsayed et al. 2021 [36]	Case report	No funding	47 years (*n* = 1)	SCS (C3–C4)	45 Hz; 450 µs pulse width; 2.8 mA right side and 3.2 mA left side amplitude	Pain decreased 80% during trial and maintained improvement through permanent implant	Improved ability to use hands	Improved ability for daily activities
Chai et al. 2017 [38]	Case report	No funding	57 years (*n* = 1)	SCS (C4 and T8)	Two leads at C4 for upper limb pain and two leads at T8 for lower limb pain	>50% pain relief during trial	NR	NR
Abd-Elsayed et al. 2015 [42]	Case report	NR	47 years (*n* = 1)	SCS (C4–C5)	NR	70–80% reduction in pain during trial with sustained relief post-implant	Improvement of function and ability to use hands	NR
Sarkar et al. 2019 [30]	Case report	No funding	55 years (*n* = 1)	SCS (C4 and T9)	10-kHz	>90% pain improvement	NR	NR
Kamdar et al. 2021 [35]	Case report	No funding	62 years (*n* = 1)	SCS (T8–T9)	60 Hz frequency; intensity of 6.5 mA	VAS pain score improved from 8/10 to 2/10 during 7-day trial that was maintained through permanent implant	Improved gait and decreased frequency of falls. Able to walk barefoot on cold surfaces and tolerated a pedicure for the first time. Sensory exam was mostly unchanged (no pinprick sensation below the mid-shin and no vibration sensation in feet)	NR
Grant et al. 2019 [31]	Case report	No funding	47 years (*n* = 1)	SCS (T9 and T10)	NR	Trial VAS score decreased to 3/10 from 7/10 at baseline; post-implant VAS: 0/10	NR	NR
Sisson et al. 2017 [32]	Case report	No funding	69 years (*n* = 1)	SCS (T9–10 disc space)	10-kHz	100% pain improvement in CPIN at 3-month follow-up	NR	NR
Panchal et al. 2016 [33]	Case report	Industrial funding	70 years (*n* = 1)	SCS (T9–T11) (Wireless staggered by 4 cm)	120 Hz and 300 µs pulse width, at an amplitude of 2.5 mA	90% improvement in pain	Able to sit, stand, walk and lay down with a significant reduction in pain	NR
Braun Filho et al. 2007 [40]	Case report	NR	72 years (*n* = 1)	SCS (T10)	80 Hz; 300 µs pulse width; 0–4 V	VAS pain score improved from 10/10 to 3/10. Pain relief was sustained 3 months after implant	NR	Improved quality of life
Wright et al. 2021 [28]	Case report	No funding	60 years (*n* = 1)	SCS (T10)	Two 8-contact dorsal column leads with intermittent burst programming	5-day trial yielded 80% pain improvement; Post-implant: VAS 0/10 was maintained at 2-year follow-up	NR	Improved quality of sleep
Abd-Elsayed et al. 2016 [29]	Case series	NR	39 years (*n* = 1)	SCS (T10–T11)	NR	95% pain relief during 1-week trial. Pain relief was sustained 3 months after implantation	NR	Improvement in sleep pattern, able to be more independent in performing daily activities
Lopes et al. 2020 [37]	Case report	NR	51 years (*n* = 1)	SCS (T10–T12)	40 Hz frequency; 350 µs pulse width; 1 V	About 50% pain improvement for both 7-day trial and permanent implant	NR	NR
Cata et al. 2004 [41]	Case report	NR	55.5 years (*n* = 2)	SCS (Patient 1 = L1, Patient 2 = T11)	Patient 1: 22 Hz; 286 µs pulse width; 0–2V. Patient 2: 80 Hz; 500 µs pulse width; 0–4 V	Patient 1: VAS pain score improved from 4.5/10 to 0.2/10 during trial and 2/10 after permanent implant. Patient 2: VAS 4.6/10 to 0/10 during trial and 3.6/10 after permanent implant	Improved gait, flexibility of legs, and touch detection for both patients. Improved sharpness detection for patient 1, none for patient 2. No change in thermal thresholds for patient 2	NR
Michael et al. 2020 [29]	Case report	No funding	48 years (*n* = 1)	SCS (NR)	NR	100% pain improvement in CIPN	NR	NR
Sayed et al. 2015 [34]	Case Report	NR	NR (*n* = 1)	SCS (NR)	NR	>50% pain relief sustained at 3-month follow-up	NR	NR
**Dorsal Root Ganglion Stimulation**								
Yelle et al. 2017 [46]	Case report	NR	49 years (*n* = 1)	DRG-S (L4–L5)	NR	>60% improvement in pain intensity	Increased walking distance without pain	Improved mood and dramatic improvement in sleep
Rao et al. 2019 [45]	Case report	No funding	53 Years (*n* = 1)	DRG-S (L5)	NR	Trial lead to >75% pain improvement	Worsened right side lower extremity numbness (buttock, posterior thigh, calf, and heel)	NR
Groenen et al. 2019 [49]	Case report	No funding	52 years (*n* = 1)	DRG-S (S1)	NR	VAS pain score improved from 8/10 to 0/10 for trial. VAS was 1/10 five months post-implantation	Regained ability to stand for prolonged period of time. EQ-5D score improved from 0.13 to 0.85. SF-36 physical component score improved from 23 to 31	SF-36 mental component score improved from 7 to 59
Finney et al. 2017 [47]	Case report	No funding	47 years (*n* = 1)	DRG-S (S1 and S2)	NR	50% pain improvement at 1-month follow-up	NR	NR
Sindhi et al. 2021 [43]	Case report	No funding	23 years (*n* = 1)	DRG-S (S3)	NR	7-day trial led to >65% pain improvement; Post-implant: >60% for 5 months	NR	Able to work 12 h shift with no pain
Kim et al. 2020 [44]	Case report	No funding	50 years (*n* = 1)	DRG-S (NR)	NR	Trial led to 100% pain improvement; Post-implant: 100% pain improvement at 3-year follow-up	NR	Able to wear shoes and exercise regularly
Grabnar et al. 2021 [48]	Case report	No funding	50 years (*n* = 1)	DRG-S (NR)	NR	VAS pain score improved from 8/10 to 0/10 during 7-day trial that was maintained through permanent implant	Lacked sensation to light touch and pinprick	Improved ability to wear shoes and exercise
**Peripheral Nerve Stimulation**								
Sacco et al. 2016 [27]	Retrospective chart review	No funding	60.5 years (*n* = 50)	PNS (auricular)	NR	All respondents reported at least some reduction in pain (>50% reduction in pain for 18 patients with quantitative results)	Improvement in numbness, gait, and balance	Improvement in sleep quality and activities of daily living. Only one patient reported adverse outcomes (intolerance of intermittent pulsing)

Visual Analogue Scale (VAS)—Pain rating scale (lower score = lower pain); EuroQol- 5 Dimension (EQ-5D)—Health-related quality of life measurement (higher score = higher quality of life); 36-Item Short Form Survey (SF-36)—Health-related quality of life measurement (higher score = higher quality of life); NR: not reported; SCS: spinal cord stimulation; DRG-S: dorsal root ganglion stimulation; PNS: peripheral nerve stimulation; C: cervical level; T; thoracic level; L: lumbar level; S: sacral level; V: volt; Hz: hertz; kHz: kilohertz; mA: milliampere.

**Table 2 biomedicines-10-01909-t002:** **Evidence Profile Table.** Evidence profile table evaluating domains per the Grading of Recommendations, Assessment, Development, and Evaluations (GRADE) criteria are displayed.

Participants (Studies) Follow-Up	Risk of Bias	Inconsistency	Indirectness	Imprecision	Publication Bias	GRADE Certainty of Evidence
**Pain relief**						
73 (23 studies)	Very Serious ^a^^,^^b^	Serious ^a^^,^^b^^,^^c^	Serious ^b^^,^^c^	Very serious ^a^^,^^b^^,^^c^^,d^	publication bias strongly suspected ^a^^,e,^^f^	⊕◯◯◯ Very low
**Improvement in neurological function**						
73 (23 studies)	Very Serious ^a^^,^^b^^,f^	Very Serious ^b^^,^^c^^,f^	Very Serious ^b^^,^^c^^,^^g^	Very Serious ^d^^,f^	publication bias strongly suspected ^e,f^	⊕◯◯◯ Very low

^a^ Included studies consisted of 22 case reports and 1 retrospective review. There was a high risk of bias in patient selection, comparability, and assessment of outcomes. ^b^ High heterogeneity was present in between and within the studies. ^c^ Some studies used dorsal column spinal cord stimulator, and some studies used dorsal root ganglion stimulator. Variations in the targeted location of chemotherapy-induced peripheral neuropathy. ^d^ Success rates widely varied between the studies. ^e^ 12/23 studies were case reports presented in conferences. ^f^ Studies reported on different neurological function outcomes. There was no consistency in the functional testing scale used. “⨁” indicates very low certainty, “⨁⨁” indicates low certainty, “⨁⨁⨁” indicates moderate certainty, and “⨁⨁⨁⨁” indicates high certainty.

**Table 3 biomedicines-10-01909-t003:** **Summary of Findings Table.** Per the Grading of Recommendations, Assessment, Development, and Evaluations (GRADE) criteria, this summary of findings table displays the certainty of evidence for each outcome of interest. Population: Cancer patients with CIPN. Intervention: SCS, DRG-S, and/or PNS trial/implant. Comparison: baseline pain/neurological function.

Outcomes	№ of Patients (Studies)	Certainty of the Evidence (GRADE)	Comments
Pain relief	73 patients (23 studies)	⊕◯◯◯ Very low ^a,b,c,d,e^	All studies reported >50% pain relief after SCS/DRG implantation. A total of 14 of 23 studies reported >70% pain relief after SCS/DRG implantation.
Neurological function	73 patients (23 studies)	⊕◯◯◯ Very low ^a,b,c,d,e,f^	Only 10 studies assessed neurological function. Of these, 6 of 10 studies reported improved gait after neuromodulation. Two studies reported improved hand motor function. Four studies reported improved sensory thresholds. Only one study reported worsening lower extremity numbness.

^a^ Included studies consisted of 22 case reports and 1 retrospective review. There was a high risk of bias in patient selection, comparability, and assessment of outcomes. ^b^ High heterogeneity was present in between and within the studies. ^c^ Some studies used dorsal column SCS, some studies used DRG-S, and some studies used PNS. There was variation in the targeted location of CIPN. ^d^ Success rates widely varied between the studies. ^e^ 12/23 studies were case reports presented in conferences. ^f^ Studies reported on different neurological function outcomes. There was no consistency in the functional testing scale used. GRADE Working Group grades of evidence - Very low certainty: we have very little confidence in the effect estimate; the true effect is likely to be substantially different from the estimate of effect. “⨁” indicates very low certainty, “⨁⨁” indicates low certainty, “⨁⨁⨁” indicates moderate certainty, and “⨁⨁⨁⨁” indicates high certainty.

## Data Availability

Data is available upon request to the corresponding author.

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
