# Peer review of "Neuromodulation Therapy for Chemotherapy-Induced Peripheral Neuropathy: A Systematic Review"

_biomedicines, 2022, doi:10.3390/biomedicines10081909_

Round 1

Reviewer 1 Report

The presented review article addresses the clinically important topic of post-chemotherapy disorders. However, the conclusions resulting from the analysis of the literature are unfortunately ambiguous.

Author Response

1) The presented review article addresses the clinically important topic of post-chemotherapy disorders. However, the conclusions resulting from the analysis of the literature are unfortunately ambiguous.

Author Response: Firstly, we would like to forward the reviewer our utmost appreciation for the time spent in reviewing this paper. We amended the conclusion section for clarity as follows: “Our systematic review performed an evidence synthesis on neuromodulation interventions for treatment of CIPN. The current evidence suggests that neuromodulation interventions, including SCS, DRG-S, and PNS, may lead to clinically meaningful pain relief in patients with CIPN. Improvement in gait, motor function, and sensory function was also highlighted in several studies. The GRADE certainty of evidence is limited due to lack of prospective comparative studies, clinical and methodological heterogeneity, and low sample size.”

Reviewer 2 Report

Dear Authors,

Thank you for this nice review of literature, 

I have one remark: the quality of evidence is scored very low, however the pain decrease is promissing,

one should emphasize that the overall succes of the now available treatment paradigms is low and more importantly the NNT and the side effects are very serious in the treatment paradigms. The treatment effects of 'conventional treatment' should be described more extensively, as their level of evidence

Neuromodulation might be a very promising technique in reducing the side effects and improve treatment efficacy. 

I guess one should explain more why the quality of evidence is low. Placebo controlled trials are very difficult in neuromodulation and so evidence will decrease,

further then that it's a nice review,

Author Response

1) Dear Authors, Thank you for this nice review of literature. I have one remark: the quality of evidence is scored very low, however the pain decrease is promising. One should emphasize that the overall success of the now available treatment paradigms is low and more importantly the NNT and the side effects are very serious in the treatment paradigms. The treatment effects of 'conventional treatment' should be described more extensively, as their level of evidence. Neuromodulation might be a very promising technique in reducing the side effects and improve treatment efficacy. 

Author Response: Firstly, we would like to forward the reviewer our utmost appreciation for the time spent in reviewing this paper. We agree with this comment that the pain decrease from neuromodulation is promising and warrants further emphasis. We amended the conclusion section to state: “The current evidence suggests that neuromodulation interventions, including SCS, DRG-S, and PNS, may lead to clinically meaningful pain relief in patients with CIPN. Improvement in gait, motor function, and sensory function was also highlighted in several studies.”

We also amended the beginning of the discussion section by adding: “Overall, this review highlights that dorsal column SCS, DRG-S, and PNS are associated with reduction in pain severity from CIPN.”

Finally, to highlight the poor success from traditional/conventional pharmacotherapy for CIPN, we added the following paragraph: “Conventional treatment options for CIPN are highlighted by the American Society of Clinical Oncology (ASCO) guidelines.42 The ASCO guidelines confirmed that no pharmacotherapeutic agents are recommended for CIPN prevention. Certain strategies such as dose delaying, dose reduction, substitution, and stoppage of chemotherapy may be considered in patients who develop intolerable symptoms from CIPN. In terms of treatment after occurrence of CIPN, duloxetine is the only current neuropathic agent with intermediate level evidence to support use in CIPN with moderate-level efficacy. Importantly, the ASCO guidelines recommended against the use of other commonly used neuropathic agents such as amitriptyline, gabapentin, pregabalin, and venlafaxine due to no evidence of efficacy or low-level efficacy. Importantly, many neuropathic medications may have a high prevalence of side effects such as somnolence, dizziness, difficulty in concentration, and other drug-specific side effects.7 Further evidence is warranted for other conservative treatment options such as scrambler therapy, acupuncture, and exercise.42

2) I guess one should explain more why the quality of evidence is low. Placebo controlled trials are very difficult in neuromodulation and so evidence will decrease,

Author Response: We thank the reviewer for this suggestion. We added the following to the limitations section: “Another reason for low-quality evidence was the absence of placebo-controlled randomized controlled trials. A prior review on placebo-controlled trials highlights improvement in neuropathic pain syndromes from neuromodulation interventions.50 However, challenges to performing placebo-controlled studies in neuromoulation are prominent and include difficulty in developing placebo or sham arms, issues with inadequate blinding, placebo and nocebo effects, and most importantly ethical concerns centered over the concept of equipoise.51

3) Further then that it's a nice review,

Author Response: We thank the reviewer for these compliments. We believe this review is now improved from addressing these suggestions from the reviewers.

Reviewer 3 Report

This study conducted by Ryan S. D’Souza et al. is interesting in that it can help patients suffering from CIPN to have an alternative treatment method than conventionally used drugs. Also, it could help basic science researchers as the study can help to better understand the treatment of CIPN. However, more explaination is needed, especially in regards to the method (i.e. nerve stimulation) used in the included studies and the underlying mechanisms of action of nerve stimulation induced analgesia. Below are my few comments: 

1) In the manuscript there should be a paragraph dedicated to the explanation of SCS, PNS, DRG. What it is, and how the stimulation is done in general. In addition, in clinic, based in what criteria the selection of the stimulation area (PN, SC, DRG) is selected? Also, is there an animal study conducted with nerve stimulation on CIPN rodents? If it is so, this should also be mention in the manuscript. 

2)  If the results of nerve stimulation is due to the gate control theory as mentioned by the authors, the analgesic effects appears to last too long. Maybe other pathways are involved in the analgesic effect of nerve stimulation that have more importance. 

3) Does the stimulation settings could have been an important factor? Please explain in the discussion part.

Author Response

1) This study conducted by Ryan S. D’Souza et al. is interesting in that it can help patients suffering from CIPN to have an alternative treatment method than conventionally used drugs. Also, it could help basic science researchers as the study can help to better understand the treatment of CIPN. However, more explaination is needed, especially in regards to the method (i.e. nerve stimulation) used in the included studies and the underlying mechanisms of action of nerve stimulation induced analgesia.

Author Response: Firstly, we would like to forward the reviewer our utmost appreciation for the time spent in reviewing this paper. Please refer to specific comments below stating how we included details on the method of neurostimulation and the underlying mechanisms of action.

2) Below are my few comments: In the manuscript there should be a paragraph dedicated to the explanation of SCS, PNS, DRG. What it is, and how the stimulation is done in general. In addition, in clinic, based in what criteria the selection of the stimulation area (PN, SC, DRG) is selected? Also, is there an animal study conducted with nerve stimulation on CIPN rodents? If it is so, this should also be mention in the manuscript.

Author Response: We thank the reviewer for this important feedback. We added two new paragraphs (Page 2, first two paragraphs) in the introduction section that is dedicated to description of SCS, DRG-S, and PNS and patient selection.

There are also animal studies that describe nerve stimulation in rodents with CIPN (e.g. Sivanesan E et al, Pain Rep. 2019. 4(5):e785). We reported findings from this study in the introduction as follows: “Studies have also highlighted that animal models with CIPN may benefit from stimulation. For instance, a study on rats with paclitaxel-induced mechanical and cold hypersensitivity revealed that SCS therapy significantly inhibited the development of mechanical and cold hypersensitivity compared to rats receiving paclitaxel alone or sham SCS.18

3) If the results of nerve stimulation is due to the gate control theory as mentioned by the authors, the analgesic effects appears to last too long. Maybe other pathways are involved in the analgesic effect of nerve stimulation that have more importance.

Author Response: We thank the reviewer for this suggestion. We added the following sentence to the discussion to describe other potential mechanisms of action: “Besides the gate control theory, analgesia from SCS may also be explained by its modulation of neurotransmitters (cholinergic, serotonergic, and opioidergic pathways), depression of wide dynamic range neurons, and activation of supraspinal levels from orthodromic dorsal column action potentials.53

4) Does the stimulation settings could have been an important factor? Please explain in the discussion part.

Author Response: We thank the reviewer for this suggestion. We added the following to the discussion section: “Although efficacy may be dependent on type of waveform as demonstrated in prior studies,59; 60 included studies did not compare efficacy between different waveforms for CIPN. Thus, comparative studies assessing different waveforms are warranted.”

Reviewer 4 Report

D’Souza and colleagues in this study titled “Neuromodulation Therapy for Chemotherapy-induced Peripheral Neuropathy: A Systematic Review” report a systematic review to assess change in pain intensity and neurological function after implementation of any neuromodulation intervention for CIPN. There are a few concerns about some of the conclusions and interpretations.

1. I would personally suggest to re-arrange Table 1, the summary should be sorted by Type of Intervention and Location which can give the readers a clearer point of view.

2. The Bias Assessment in Results section is confusing. What is “2 stars” and “no stars” mean? Table 2 is unnecessary.

3. Table 3 and 4 need to be re-arranged to make them readable. Footnote should be used to explain some symbol.

4. The discussion section needs to be improved instead of repeating the findings in results section. The authors should add more constructive recommendations to the clinicians.

Author Response

1) D’Souza and colleagues in this study titled “Neuromodulation Therapy for Chemotherapy-induced Peripheral Neuropathy: A Systematic Review” report a systematic review to assess change in pain intensity and neurological function after implementation of any neuromodulation intervention for CIPN. There are a few concerns about some of the conclusions and interpretations.

Author Response: Firstly, we would like to forward the reviewer our utmost appreciation for the time spent in reviewing this paper. Please see our responses below to each suggestion.

2) I would personally suggest to re-arrange Table 1, the summary should be sorted by Type of Intervention and Location which can give the readers a clearer point of view.

Author Response: We thank the reviewer for this suggestion. We re-arranged Table 1 into separate subsections titled by type of intervention (SCS, DRG-S, and PNS). We also arranged the rows in the order of descending spinal level.

3) The Bias Assessment in Results section is confusing. What is “2 stars” and “no stars” mean? Table 2 is unnecessary.

Author Response: We thank the reviewer for this feedback. To clarify what the stars imply in the bias assessment from the Newcastle-Ottawa scale, we added the following to the methods section: “Higher number of stars indicates a lower risk of bias for each respective domain.”

We removed Table 2 from the manuscript and described the risk of bias in the results text as follows: “Using the Newcastle-Ottawa Quality Assessment Scale, the study by Sacco et al19 received two stars for the Selection domain (maximum possible four stars), and two stars for the Exposure/Outcome domain (maximum possible three stars). There were no stars assigned to the Comparability domain, indicating high risk of bias for that respective domain.”

4) Table 3 and 4 need to be re-arranged to make them readable. Footnote should be used to explain some symbol.

Author Response: We have re-arranged Tables 3 and 4 (which are now Table 2 and 3 in the revised manuscript) to make them more readable. The footnotes were clarified and expanded in the revised manuscript in the respective table legend.

5) The discussion section needs to be improved instead of repeating the findings in results section. The authors should add more constructive recommendations to the clinicians.

Author Response: We thank the reviewer for this important suggestion. We have completely revised the discussion section to remove repetitive information and to focus on constructive recommendations for clinicians. We added three new paragraphs to the discussion section to address this reviewer suggestion.

Round 2

Reviewer 2 Report

Thank you for the changes

Reviewer 3 Report

The authors have clearly answered to all my questions. 

Reviewer 4 Report

The authors addressed all my concerns.